# High-Value Plant Species Used for the Treatment of “Fever” by the Karen Hill Tribe People

**DOI:** 10.3390/antibiotics9050220

**Published:** 2020-04-29

**Authors:** Methee Phumthum, Nicholas J. Sadgrove

**Affiliations:** 1Department of Pharmaceutical Botany, Faculty of Pharmacy, Mahidol University, Bangkok 10400, Thailand; 2Sireeruckhacharti Nature Learning Park, Mahidol University, Nakhon Pathom 73170, Thailand; 3Jodrell Science Laboratory, Royal Botanic Gardens Kew, Richmond, London TW9 3AE, UK

**Keywords:** medicinal plants, antipyretic, ethnobotany, Thai, ethnic group, infection, antiviral, antiprotozoal

## Abstract

The symptom “fever” is generally not itself a terminal condition. However, it does occur with common mild to severe ailments afflicting the world population. Several allopathic medicines are available to attenuate fever by targeting the pathogen or the symptom itself. However, many people in marginal civilizations are obligated to use locally grown medicinal plants due to limited access to common pharmaceuticals. The Karen ethnic group is the biggest ethnic minority group in the hill-tribes of Thailand. They utilise a vast repertoire of medicinal plant species. Since many modern drugs were discovered out of traditional therapies, it is possible to discover new allopathic drugs in the treatment of fever and associated pathogens from the Karen people. Thus, this study aims to identify and record the ethnomedicinal plants they used for the treatment of “fever”. The names of plants used by the Thai Karen people for the treatment of fever were mined from publications on ethnomedicinal uses. Useful plant species and families were identified using the Cultural Importance Index (CI). With the mined data, 125 plant species from 52 families were identified, distributed across 25 Karen villages. A chemical cross-examination of these species provided valuable insights into chemical classes worthy of further investigation in the context of fever and associated pathogens.

## 1. Introduction

Fever is one of the most prevalent afflictions of the human race in the developing world [1]. There are several infectious diseases that involve fever as a symptom. Fever commonly develops in the course of microbial infection, where pyrogens released by the bacteria or virus act directly on the OVLT (organum vasculosum laminae terminalis) of the anterior hypothalamus, which responds by elevating core body temperature [2]. However, the endogenous release of pyrogens occurs in response to pathogenic protozoal lifeforms that use mosquitoes as their vectors. In northern Thailand, the most common forms of fever are caused by the transmission of mosquito-borne vectors, particularly dengue fever [3] and malaria [4]. The recent global pandemic of COVID-19 has stimulated a renewed interest in discovering new therapies that act against fevers and fever-causing agents. Seeking curative drugs from plants is considered a rational approach because of recent developments in the context of synthetic derivatives of alkaloids from Cinchona, possibly contributing to patient recovery [5].

While the true function of fever in disease recovery is unknown, it is conceived that fever is part of an infection-fighting mechanism wherein the host’s core body temperature increases to the point that stresses the pathogenic microbe [6,7]. Although fever is not itself regarded as life-threatening, it creates enough discomfort to become a hindrance to normal activities in society, leading to absences from places of work or study. The statistics convey that societies spend a lot of money on doctors and at health centres, paying for prescribed treatments.

Medicinal plants have been utilized in human civilizations since time immemorial [8]. The utilization continues today in select populations of developing countries, with near-total dependence in some cases (primary healthcare). In less remote areas, plant-based therapies are still being used as optional alternatives to allopathic drugs [9]. Preserving the knowledge of these alternative therapies is necessary because history demonstrates that modern medicines can be developed from plants used ethnomedicinally [10]. For example, the discovery of artemisinin aided in the recovery of millions of people from around the world infected with *Plasmodium falciparum*. The discoverer, Tu Youyou, won half of the Nobel Prize in 2015 [11].

Thailand boasts over 10,000 higher plants [12] and a variety of ethnic groups [13,14], making it a hot-spot for drug discovery, guided by medicinal plant selection. Specifically, the Karen people are an ethnic group that settled in Thailand after migration from Myanmar in the 18th century during a time of war and unrest [6,7]. Today approximately 55,000 Karen people are found living in 1930 villages that occupy 16 provinces in northern and western Thailand [15]. Their villages are located near the forests within proximity to a stream to feed their swidden-based farming [16]. Almost all of the Karen people adhere to animism, which connotes a belief that spirits are omnipresent and are categorized according to location or family ties, such as ancestral spirits, house spirits, forest spirits, farm spirits, land spirits, and so forth [17]. Because of their utter dependence on natural resources, the Karen people have immense knowledge of medicinal plants [18]. Therefore, the Karen villages are a treasury of ethnomedicinal plant knowledge.

Individual medicinal plants are frequently reported to have multiple uses. For example, one medicinal plant may be therapeutic against two or more ailments. In 2018 it was demonstrated that there are over 16,000 medicinal uses for 2187 plant species in Thailand. Furthermore, with more ethnic groups to be interviewed, it was predicted that the total number of medicinal plants is actually higher [19].

The study of ethnobotany was only introduced to the Thai academic system in 1990 [20]. The oldest thesis that meets the standard of the current Thai ethnomedicine paradigm was published in 1992. It reported 473 uses from just 133 species. However, at that time, only 36 species were described as a treatment for fever or the common cold [21]. Soon after this, a master′s thesis that focused on plants utilized for various medical purposes by the Karen people in Chiang Mai listed only 43 plant species [22]. However, the momentum of ethnobotanical studies appearing in the literature began to increase steadily. Most of the ethnobotanical publications coming out of Thailand were published in the last 10 years. It was not until 2014 that plants specific to fever became the focus of studies, with the first report describing 83 plants used by people living in Krabi [23].

Fever as a focus for ethnomedicinal studies in the villages of northern Thailand makes sense because dengue fever is a common ailment [3]. Another common infection associated with fever, specifically in the hill-tribe youth, is hepatitis B, which particularly afflicts the Karen people [24]. The Karen people are also susceptible to malaria [4], and skin and upper respiratory tract infections [25]. Thus, fever is common to the plight of the Karen people.

The oldest ethnobotanical study to focus specifically on the Karen people was published as a book in 1993 [26]. Due to the extreme remoteness of the Karen people, their societies became a very attractive place for ethnobotanical discoveries. Thus, many students wrote their masters and PhD theses on ethnomedicinal uses by the Karen people. Most of the studies were based in Chiang Mai province, Northern Thailand. Strangely, even with the obvious importance of therapies that target fever, there are still no studies that focus exclusively on the Karen people’s techniques for treating this symptom.

Inspired by the lack of a coherent summary of plants that target fever in the tradition of the Karen people, this study aims to build an exhaustive list of plants used by the Karen people for fever. Specifically, the study proposes to answer the following questions: (1) How many plant species and families were used for the treatment of fever by the Karen people in Thailand? (2) What are the names of the plant species that demonstrate the highest value for the treatment of fever and could be flagged as having value in a research and development setting that seeks to identify drugs of interest to the wider pharmaceutical industry? 

## 2. Results

Out of the 25 Karen villages (Figure 1), 158 use-reports from 125 species in 52 families were recorded. Leguminosae had the highest numbers of use-reports (18) with a CI value of 0.72. Asteraceae and Acanthaceae also had high CI values from the use-reports, with the values of 0.44 and 0.40, respectively. Rutaceae (9 use-reports) and Lamiaceae (8 use-reports) had CI values slightly greater than 0.3. Apocynaceae and Euphorbiaceae had a similar CI value (0.24), which is slightly higher than Malvaceae (0.20; see Figure 2 and Appendix A).

At the species level, *Mimosa pudica* L. had the highest CI value (0.16). The CI value of *Acorus calamus* L., *Elephantopus scaber* L., *Melicope glomerata* (W. G. Craib) T.G. Hartley, *Phyllanthus amarus* Schumach. & Thonn., and *Scoparia dulcis* L. were all equal to 0.12. Twenty other plants had CI values of 0.08, and the remainder of the species were cited only once for curing fever (Appendix A).

## 3. Discussion

### 3.1. Ethnobotanical Significance

The Karen are the biggest hill-tribe ethnic group in Thailand. It is, therefore, not surprising that the current study successfully compiled the biggest list of medicinal plants used for fever treatment in Thailand. The previous survey of medicinal practice in Krabi province of southern Thailand reported 83 plant species from 47 plant families used to treat fever [23]. However, the Karen people of the current study use 125 plant species from 52 plant families for the same purpose. To build such a comprehensive dataset required the mining of ethnobotanical information and conducting semi-structured interviews across more Karen villages than in any other study so far. Previous studies were generally much smaller, with up to four Karen villages visited in Chiang Mai province [22,27]. This excludes the species distribution model (SDM) that used metadata from 12 Karen villages in Chiang Mai [28]. In the current study, the synthesis of published species lists and fieldwork data across 25 Karen villages covers almost all ecological regions that the 2000 or so Karen villages occupy in both northern and western Thailand (Figure 1).

Out of the 52 plant families identified, the Cultural Important Index (CI) values indicated that Leguminosae is the most important in the context of fever treatment. This is mirrored in its high frequency of use as a food, which is about 20% more frequent as compared to medicinal plant use [29]. The remaining important families, from highest to lowest, are Asteraceae, Acanthaceae, Rutaceae, Lamiaceae, Apocynaceae, Euphorbiaceae, and Malvaceae. A study of medicinal plants that target fever, used by the Krabi people (Southern Thailand), agreed that Leguminosae comprised the highest number of species, but Euphorbiaceae was also highly cited, which is strikingly different to northern Thailand tribes [23]. Nevertheless, the eight families identified in the current study are also widely recognised for medicinal effects other than fever attenuation. Among the 206 Thai ethnomedicinal plant families, these eight specifically demonstrated high family use values (UVf), which is [19] the CI value but calculated in different ways. Moreover, aside from being estimated as important medicinal plant families in Thailand [30], they have received this recognition in other parts of the world [31,32,33,34].

While this study covered many Karen villages and compiled a comprehensive list of species, the number is still smaller than the total of medicinal plants used for fever treatment in the country. There are several species that did not appear in Karen materia medica for fever that are reportedly used for this purpose by people in southern Thailand and other ethnic groups living in northern Thailand (i.e., Hmong, Mien, Khamu, and Lua). Alternatively, the Karen people also reported species names that were not used by the other groups [20,23]. Several factors could explain this, which may include the face of exclusivity of traditional knowledge across ethnic groups [35]. Another explanation could be the chemovariation of plant species, or heterogeneous species aggregates requiring taxonomic revision.

Despite wide coverage of geographical provenances and vegetative zones in the current study, there are almost 2000 Karen villages in Thailand [15]. A study of ethnomedicinal plant diversity in Thailand found that the listed 2187 Thai ethnomedicinal species that were used by people in 121 villages; evidently, this does not represent all ethnomedicinal flora used across the entire of the country. To create a more comprehensive list, it is necessary to carry on with fieldwork, focusing on other Karen villages to compile more ethnomedicinal data. There are still many unexamined areas, especially in the southern parts of the northern region and the western region (Figure 1).

### 3.2. Ethnopharmacological Significance

Therapies that attenuate or resolve fever are mechanistically diverse. Some therapies target the pathogen while others target the symptom. For example, quinoline-containing drugs, such as chloroquine and quinine, resolve fever by acting with high selectivity against the lysosomes of malaria trophozoites [36]. Recent evidence supports the use of quinoline-based drug designs in the treatment of coronaviruses, such as the strain responsible for the SARS epidemic of 2002–2003 [37]. Artemisinin also targets *Plasmodium falciparum* with high selectivity in a heme-mediated degradation of the peroxide bridge [38].

Alternatively, some therapies target the fever itself via inhibition of the release of the specific pyrogens that communicate with the hypothalamus to bring about elevated body temperature [39]. Antipyretics follow a diverse number of pathways, but the most common example to illustrate this mechanism is aspirin, a derivative of salicin from willow bark (*Salix* spp.), which inhibits COX-2 then reduces the levels of PGE(2) in the hypothalamus [39].

The Karen people are susceptible to a range of pathologies associated with fever, including pathogens of both viral and protozoan denomination. Thus, the species with the highest CI values in the current study may function either by pathogenic mediation or as antipyretics. Furthermore, many of the ingested medicinal species used in traditional practice derive therapeutic effects from metabolic forms of the natural plant-derived prodrug [40], such as in the case of salicin, which is metabolically converted to salicylic acid [39].

The most cited species for fever treatment in the tradition of the Karen people was *Mimosa pudica* L. from Leguminosae. This species is also traditionally used for the treatment of fever in Cambodia [41]. The plant produces several classes of compounds, such as terpenoids, flavonoids, glycosides, and alkaloids [42]. The Leguminosae family is well known for its antibacterial flavonoids and alkaloids. The presence of enzyme-inhibitory amines in *M. pudica* provides an initial clue into the possible mechanism of this species. The unique amino acid *L*-mimosine (Figure 3a) may be the best starting point for further investigation into the efficacy of this species [43], examining for antipyretic activity and pathogen mediation. An extract of this species has already demonstrated antiviral activity against the mumps virus [44].

Out of the species with the highest CI values, *M. pudica* (Leguminosae), *Melicope glomerata* (Rutaceae), and *Elephantopus scaber* (Asteraceae) were the only three that also belong to families with high CI values. Chemical studies of *E. scaber* have isolated and identified several unique compounds from the tuber, including sesquiterpene lactones, phenolic amides [45], and a very uncommon lignan, which is a δ-truxinate derivative (3,4,3′,4′-tetrahydroxy-δ-truxinate: (Figure 3b). It was predicted *in silico* to confer anti-inflammatory effects [46], but no studies have focused on activities specific to pathogens. The unusual chemistry of this species justifies further investigation.

Unlike *Elephanthopus scaber* and *Mimosa pudica*, there is no detailed knowledge of the chemistry of *Melicope glomerata*. Other species of *Melicope* quinoline alkaloids (the same group as quinine/chloroquine) have been identified [47,48], but with a similar cyclobutane moiety as the δ-truxinate (melicodenine D: (Figure 3c), so it is reasonable to expect related structures in *Melicope glomerata*. With the prevalence of resistance to quinine across the world [49] and predicted resistance development to artemisinin [38], *Melicope glomerata* may be a valuable research option in the search for alternatives.

Although it is a well-known species in the *materia medica* of Thailand, *Andrographis paniculata* (Burm.f.) Nees from the Acanthaceae family had a low CI value as a fever plant, according to Karen ethnomedicinal uses. However, in other parts of the world *A. paniculata* is highly regarded for attenuation of cold- and influenza-related afflictions [50]. This species has already been demonstrated in vitro as having the potential for fever treatment. It produces the labdane andrographolide (Figure 3d), which is a lipophilic diterpene that potentially mediates its antipyretic effects via COX-2 inhibition, which sees an overall reduction in PGE(2) in the hypothalamus region [51]. A related species that is also used to treat fever, *A. lineata* [52], yielded a near-identical δ-truxinate derivative (dimethyl 3,4,3′,4′-tetrahydroxy-δ-truxinate: (Figure 3e)) to that found in *E. scaber*, as mentioned above, differing only by homology of one of the esters [53]. This pattern of truxinate derivatives highlights the potential value in examining this chemical species further.

Three of the species with high CI values are not members of any of the families with high CI values. These are *Acorus calamus* (Acoraceae), *Phyllanthus amarus* (Phyllanthaceae), and *Scoparia dulcis* (Scrophulariaceae). A chemical study of *A. calamus* isolated acoradin (Figure 3f), a lignan similar to a truxinate [54], which again highlights the necessity to investigate this chemical class. A series of five more noteworthy anti-inflammatory lignans, hypophyllanthin: (Figure 3g), can be derived from *P. amarus* [55]. Since the extract of this species is a highly regarded treatment for hepatitis B [56], these lignans may be involved in antiviral effects. Lastly, the introduced species *S. dulcis* accumulates a series of scopadulane-type meroditerpenes scopadulcic acid B: (Figure 3h) with an unusual benzoic ester moiety [57]. These derivatives demonstrated antiviral activity against herpes simplex virus type-1 [58]. Thus, it is necessary to investigate the scopadulane meroditerpenes in the context of fever.

Another of the lesser mentioned species for fever treatment but with widely known therapeutic attributes is *Plantago major* L. from the Plantaginaceae family. Due to its popularity in other parts of the world, some research has been done on the species. A tentative study describes in vitro antipyretic activity in mice [59]. It has also been demonstrated to have antiviral effects [60]. While this species has well-known sterols and triterpenes, phenolics and tannins, a few of the structures that are unique to this species include a plethora of iridoids and two alkaloids, indicain and plantagonin (Figure 3i) [60]. The iridoids and alkaloids are likely to be of the most interest in examining therapeutic effects in vitro.

## 4. Materials and Methods

Names of medicinal plants used in the treatment of fever by the Karen tribes in Thailand were collected from published records. The period of available publications ranged from 1993 to 2019. The main source of data was the compilation of works by students in the form of theses, which are available in university libraries and the website of the Thai Library Integrated System (www.tdc.thailis.or.th). The web site also provides scientific reports from all higher educational institutes in Thailand.

More data was garnished from publications that were identified by searching online databases, namely Scopus, Google Scholar, and PubMed. This included journal articles published only in Thai journals, wherein a reiteration of postgraduate theses was occasionally observed. If it could be confirmed that a published article contained the same data that we had previously recorded from an author’s earlier work in a thesis format, the data was treated as a single entry for both appearances. Another measure to avoid repetition of data was to exclude studies that used metadata for analysis. Furthermore, plant species with no full binomial Latin name specified (for example, *Cinnamomum* sp.) were excluded. Finally, all Latin names were updated following The Plant List database (www.theplantlist.org).

Thus, data that was included in the study derived from a total of 16 sources, which included 14 theses, one journal article, and one book (see Appendix A). From published data alone, ethnomedicinal uses covered 25 villages (Figure 1).

### Plants with High Potential

Plant species with higher potential as efficacious treatments were identified based on the frequency of ethnomedicinal reports. The modified Cultural Important Index (CI) [61] was applied to identify these species. For the index to be calculated, it first requires the determination of two factors. One factor is the total number of use-reports (UR) of a species, which is the number of mentioned uses of a species by all informants. The other is the total number of informants, which is denoted by N. The index is calculated using the following equation:CI = UR/N(1)

Since the one species can have multiple uses, the CI value of a species can be above 1, but in the current study, we were only assessing the one use, as a treatment for fever, so the value for a single species is always between 0 and 1. Summed values to represent plant families can be higher (but in the current study, no values exceeded 1). The lowest possible CI value is zero (0), which indicates the plant is not useful for the treatment and was not mentioned by any of the informants. Alternatively, the higher the CI value, the more interesting the species is, which may reflect the measure of its efficacy. It also implies that the species is a point of interest for research and development. Some studies might prefer to use the other index as “use value (UV)” to identify important plant taxa because the CI and UV values are equivalent, but they are calculated differently [61].

Unfortunately, as we used available data from previous studies, we could not specify the exact number of informants from each study. Therefore, we use N to denote “pseudo-informant” in the current study, which represents a studied village or a data source that reported ethnomedicinal uses, instead of a person who is giving ethnomedicinal data [19].

## 5. Conclusions

The current study has compiled the most comprehensive list to date of botanical species that are treated as therapies against fever by the Karen people. The study covered 25 Karen villages in Thailand and compiled a list that includes 125 species. The most cited families in order from highest to lowest were Leguminosae, Asteraceae, Acanthaceae, Rutaceae, Lamiaceae, Apocynaceae, Euphorbiaceae, and Malvaceae. Several species that were mentioned consistently across the many villages were *Mimosa pudica* L., *Acorus calamus* L., *Elephantopus scaber* L., *Melicope glomerata* (W.G. Craib) T.G. Hartley, *Phyllanthus amarus* Schumach. & Thonn., and *Scoparia dulcis* L. By cross-examination of the chemical character of these species, it was demonstrated that an unusual lignan with a cyclobutane moiety was evident in many of the compound structures. A series of quinoline alkaloids also had this cyclobutane moiety. Other lignans were also observed, as well as two interesting lipophilic diterpenes that could be involved in COX-2 inhibition. The structures identified in the current study are highlighted as potential candidates in research on anti-fever therapeutics via antipyretic and antipathogenic mechanisms. 

## Figures and Tables

**Figure 1 antibiotics-09-00220-f001:**
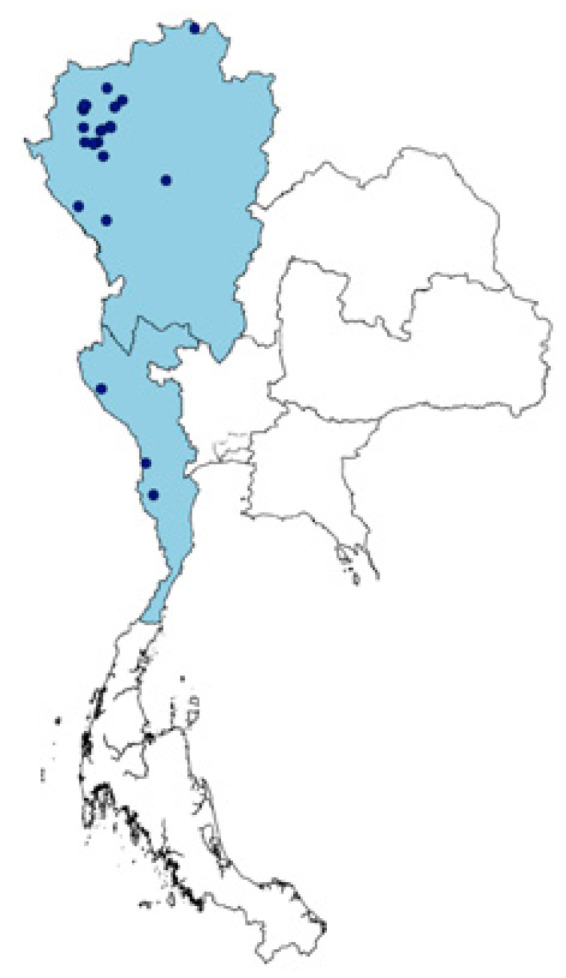
A map of Thailand shows regions that Karen are occupying (blue areas) and Karen villages that have available ethnomedicinal data (dark blue dots).

**Figure 2 antibiotics-09-00220-f002:**
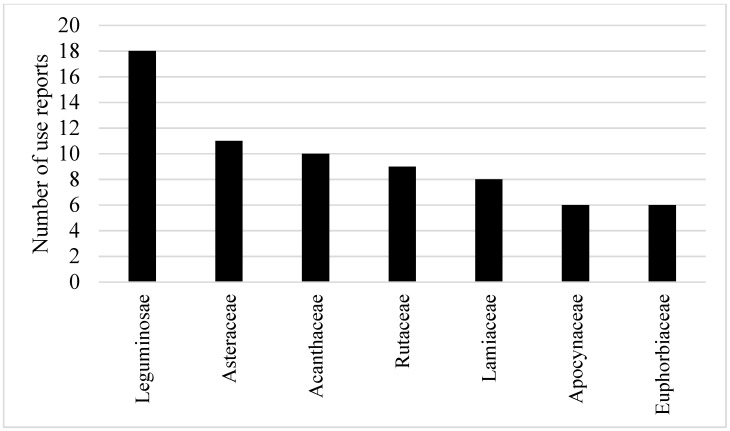
Numbers of use reports of the top seven families having the highest use report for fever treatment by 25 Karen villages in Thailand.

**Figure 3 antibiotics-09-00220-f003:**
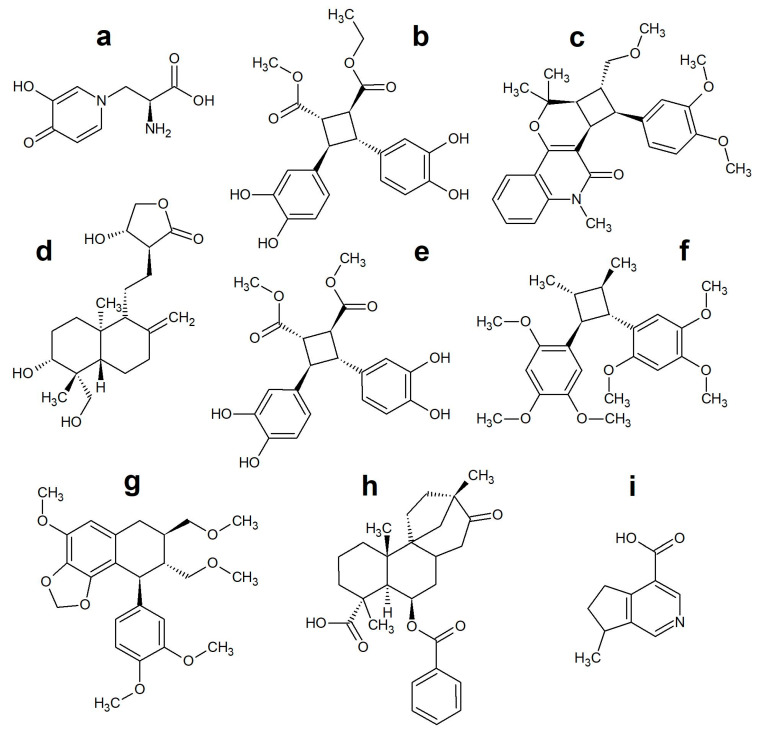
Molecular structures identified as having a potential value in the search for drugs to combat fever or associated pathogens: (**a**) L-mimosine, (**b**) 3,4,3′,4′-hydroxy-δ-truxinate, (**c**) melicodenine D, (**d**) andrographolide, (**e**) dimethyl-3,4,3′,4′-hydroxy-δ-truxinate, (**f**) acoradin, (**g**) hypophyllanthin, (**h**) scopadulcic acid B, (**i**) plantagonin.

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
