# Peer review of "High-Value Plant Species Used for the Treatment of “Fever” by the Karen Hill Tribe People"

_antibiotics, 2020, doi:10.3390/antibiotics9050220_

Round 1

Reviewer 1 Report

Methee Phumthum and co-authors build an exhaustive list of plants used by the Karen people for fever. Seeking curative drugs from plants is considered a rational approach during recent global pandemic of COVID-19. This study covered 25 Karen villages in Thailand and compiled a list that includes 125 species. Identified structures in the current study are highlighted as potential candidates in medical research. The reviewer believes the manuscript was well written, the science behind is sound and the discussion was solid, this work will attract great attentions from the scientific community of medicinal chemistry, organic synthesis and others. Therefore, the reviewer recommends the acceptance of this paper after addressing a suggestion below:

  1. Molecular structure 7 in Figure 4 should be “Hypophyllanthin” not “Hyphyllanthin”.

Author Response

Dear Reviewer,

Thank you very much for your comments.

We have changed our manuscript as the response below. 

Molecular structure 7 in Figure 4 should be “Hypophyllanthin” not “Hyphyllanthin”.

Answer– Correction was made.

Reviewer 2 Report

Manuscript entitled "High-value plant species used to treat ‘fever’ by the Karen hill-tribe people: A meta-analysis" discloses a meta-analysis of various plant species used by Karen hill-tribes from Thailand to treat common fever. This review is mainly based on collection of medicinal plant as a remedy for fever. In literature there are number of articles based on medicinal plants found at this location. Although this manuscript especially focused on fever, it needs some improvements mentioned below:

  • Since the basis of manuscript is to identify and record the ethnomedicinal plants to treat fever it is pertinent to provide recently published articles as well such as https://www.ncbi.nlm.nih.gov/pmc/articles/PMC6963713/
  • Title of the manuscript should be adjusted appropriately. Please avoid filler phrases such as “used to” What meta-analysis mentioned in the title?
  • Abstract needs to be redrafted for English language and grammar. Ex. Line 16: Karen people are the biggest ethnic group in “the hill-tribes in Thailand.” They have a vast “amount” of….
  • Introduction starts with the quote from William Osler which seems to be philosophical rather than scientific/research oriented. Authors should carefully check this section for English grammar.  
  • Line 88-89: There is no correlation between this sentence and research area covered in this manuscript so it should be avoided.
  • In results section, Authors should provide CI values for Asteraceae and Acanthaceae.
  • Authors should mention clearly that on what basis CI values were obtained as shown in figure 2 which claims based on numbers of use reports. Please clarify whether these results are based on intensity of use of entire plant or its specific parts.
  • Figure 6: compounds numbering size should be adjusted appropriately.

Author Response

Dear Reviewer,

Thank you very much for the comments. Our answers are shown below.

  • Since the basis of manuscript is to identify and record the ethnomedicinal plants to treat fever it is pertinent to provide recently published articles as well such as https://www.ncbi.nlm.nih.gov/pmc/articles/PMC6963713/
  • Answer:

    MP -The paper has been already included. Please see ‘list of data sources’ in the Supplementary materials 1.

  • Title of the manuscript should be adjusted appropriately. Please avoid filler phrases such as “used to” What meta-analysis mentioned in the title?
  • Answer

    MP -We modified the title by removing ‘meta analysis’.

    NS- The title is currently grammatically correct. ‘Used to’ is not a filler phrase, without it the title conveys poor English.

  • Abstract needs to be redrafted for English language and grammar. Ex. Line 16: Karen people are the biggest ethnic group in “the hill-tribes in Thailand.” They have a vast “amount” of….
  • Answer

    NS-Abstract has been improved.

  • Introduction starts with the quote from William Osler which seems to be philosophical rather than scientific/research oriented. Authors should carefully check this section for English grammar.
  • Answer

    MP-We rewrote the phrase.

    NS-The introduction was checked carefully for areas of improvement. 

  • Line 88-89: There is no correlation between this sentence and research area covered in this manuscript so it should be avoided.
  • Answer

    NS- There is currently a pronounced research interest in chlorquinine in the context of alleviation symptoms of COVID-19. This has awakened research interest in plant specialised metabolites in the context of antiviral therapies. As is evident from the comments of reviewer 1, this article with attract the interest of many natural products chemists seeking to find antiviral plants.

    MP -We moved the paragraph to the beginning of the Introduction.

  • In results section, Authors should provide CI values for Asteraceae and Acanthaceae.
  • Answer:

    MP -We changed the sentence from “with 11 and 10 use reports respectively.” to “with the value of 0.44 and 0.40 respectively.

  • Authors should mention clearly that on what basis CI values were obtained as shown in figure 2 which claims based on numbers of use reports.
  • Answer

    MP -We add few sentences at the Materials and Methods.

  • Please clarify whether these results are based on intensity of use of entire plant or its specific parts.
  • Answer

    MP -We totally agree that the plant parts are not related to the main content of the manuscript. So, we decided to remove the paragraph and the Figure 3.

  • Figure 6: compounds numbering size should be adjusted appropriately.
  • Answer: 

    N.S.-The numbers have been shifted to make them line up.

Reviewer 3 Report

The reviewed manuscript is an ethnobotanical study focused on anti-fewer natural remedies used Karen people in Thailand. The research is based on the review of data collected for 26 years and published in a form of 16 sources in total. Because the investigations were performed on the Karen tribes and covered 25 villages, I can suspect some uniformity in medicinal plants' natural resources as well as the knowledge on their utilisation. In my opinion, such investigated material is enough to re-elaborate in a form of meta-analysis to screening the most promising species for future investigations in treatment of diseased with the fever as a common symptom. The manuscript is clearly written and well organised, findings are discussed correctly with the proper literature cited.

I propose to accept the manuscript with minor revision, concerning mainly spelling mistakes.

Author Response

Dear Reviewer,

Thank you very much for your comments. Please consider our answers below.

I propose to accept the manuscript with minor revision, concerning mainly spelling mistakes.

Answer: 

The manuscript was scanned for spelling errors. Most changes were made in the abstract. Thank you for your comments.

Round 2

Reviewer 2 Report

Manuscript Title: High-value plant species used to treat ‘fever’ by the Karen hill-tribe people: A meta-analysis.

The authors have made an attempt to answer all the questions and corrected manuscript as suggested by Reviewers. As for the evaluation and improvement of this manuscript these corrections were important and significant. They have responded well to all suggestion and corrections. Therefore, this improved version of the manuscript can be accepted for publication upon following minor modifications.

  • Figure 2: Y axis “Number of use reported” can be changed to “ Frequency of use”
  • Figure 2: legend: “Literature reported use of various types of plant species for fever treatment by 25 Karen villages of Thailand.”
  • Please make sure that all DOI of reference list is with same color.